# Combination of Thymoquinone and Intermittent Fasting as a Treatment for Breast Cancer Implanted in Mice

**DOI:** 10.3390/plants13010035

**Published:** 2023-12-21

**Authors:** Shatha Khaled Haif, Lina T. Al Kury, Wamidh H. Talib

**Affiliations:** 1Department of Clinical Pharmacy and Therapeutics, Applied Science Private University, Amman 11931-166, Jordan; shaza-haif@hotmail.com; 2Department of Health Sciences, College of Natural and Health Sciences, Zayed University, Abu Dhabi 144534, United Arab Emirates; 3Faculty of Allied Medical Sciences, Applied Science Private University, Amman 11931-166, Jordan

**Keywords:** breast cancer, thymoquinone, intermittent fasting, MTT assay, IGF-1, β-hydroxybutyrate, leptin

## Abstract

Breast cancer stands out as a particularly challenging form of cancer to treat among various types. Traditional treatment methods have been longstanding approaches, yet their efficacy has diminished over time owing to heightened toxicity, adverse effects, and the emergence of multi-drug resistance. Nevertheless, a viable solution has emerged through the adoption of a complementary treatment strategy utilizing natural substances and the incorporation of intermittent fasting to enhance therapeutic outcomes. This study aimed to assess the anticancer activity of thymoquinone (TQ), intermittent fasting, and their combination using in vivo and in vitro methods. The anti-proliferative activity of TQ and fasting (glucose/serum restriction) were evaluated against the T47D, MDA-MB-231, and EMT6 cell lines and compared to normal cell lines (Vero) using the MTT colorimetric assay method. Additionally, this study aimed to determine the half-maximal inhibitory concentration (IC_50_) of TQ. For the in vivo experiment, the antitumor activity of TQ and intermittent fasting (IF) was assessed by measuring the tumor sizes using a digital caliper to determine the change in the tumor size and survival rates. At the molecular level, the serum levels of glucose, β-hydroxybutyrate (β-HB), leptin, and insulin growth factor-1 (IGF-1) were measured using standard kits. Additionally, the aspartate transaminase (AST), alanine transaminase (ALT), and creatinine serum levels were measured. The inhibition of the breast cancer cell lines was achieved by TQ. TQ and intermittent fasting both had an additional anticancer effect against breast tumors inoculated in mice. The combination therapy was evaluated and found to significantly reduce the tumor size, with a change in tumor size of −57.7%. Additionally, the combination of TQ and IF led to a decrease in the serum levels of glucose, IGF-1 (24.49 ng/mL) and leptin (1.77 ng/mL) while increasing β-hydroxybutyrate in the mice given combination therapy (200.86 nM) with no toxicity on the liver or kidneys. In the mice receiving combination therapy, TQ and IF treated breast cancer in an additive way without causing liver or kidney toxicity due to decreased levels of glucose, IGF-1, and leptin and increased levels of β-hydroxybutyrate. Further investigation is required to optimize the doses and determine the other possible mechanisms exhibited by the novel combination.

## 1. Introduction

Cancer ranks as one of the most prevalent global causes of death. According to 2015 estimates from the World Health Organization (WHO), it stands as the third or fourth primary cause of death before the age of 70 in 22 countries, and in 91 out of 172 countries, it takes the lead as the first or second leading cause of death [1,2]. In recent times, breast cancer has surpassed other types of cancer as the most commonly diagnosed worldwide. The latest global estimates on cancer prevalence suggest that, in 2020, there will be 2.26 million new cases of breast cancer, establishing it as the foremost cause of cancer-related mortality among women globally [3,4].

Despite significant advancements in anticancer therapy, such therapies have unfortunately limited the efficiency for tumor elimination and are very hazardous to healthy cells [5]. As a result, developing new therapeutic ways has emerged as a paramount focus in cancer research. This includes a healthy lifestyle and maintaining a healthy body weight, in addition to searching for anticancer compounds derived from natural sources. Such natural products are commonly thought to be less toxic and have fewer side effects [6].

Cancer cells are extremely susceptible to nutrient deprivation because they have a different metabolism than healthy cells. Recent animal studies have demonstrated the benefits of intermittent fasting (IF), which is described as a “calorie reduction (CR) without malnutrition” and involves a 10–40% reduction in calories compared to the controls while maintaining adequate nutrition. IF has the potential to impede the advancement of various cancers comparable to chemotherapy. Simultaneously, it can act as a safe guard for healthy cells, shielding them from the adverse effects of chemotherapy drugs, while subjecting cancer cells to the treatment [7]. Chronic CR reduces the levels of many hormones (such as leptin) [8], glucose, growth factors (such as IGF-1), and cytokines in the blood, resulting in less growth factor signaling, fewer vascular disturbances, and reduced inflammation [9]. Collectively, CR can slow the onset and progression of cancer. In fact, the alteration of IGF-1 signaling has a crucial role in the antineoplastic effects of CR, according to preclinical investigations in colon, pancreatic, and breast cancers [10].

For a very long time, breast cancer has been treated with chemotherapy, surgery, radiotherapy, and hormonal therapy. These therapeutic methods have become progressively ineffective because of drug resistance to multiple drugs and severe side effects [11]. However, since it has been demonstrated that substances derived from natural sources have significant anticancer activities, using a complementary treatment strategy could be very beneficial in this case. Such organic compounds can reduce the virulence of breast cancer, restrict the growth of cancerous cells, and modify the pathways that lead to cancer [6].

Thymoquinone (TQ) (2-methyl-5-isopropyl-1,4-benzoquinone) is a phytochemical compound found in black cumin (*Nigella sativa*) (NS) with a long history of medicinal use, which with the appropriate modifications and additional clinical research, could be very effective against breast cancer [12]. TQ is the most abundant component in the volatile oil of NS seeds, and it is responsible for the majority of the herb’s properties. TQ has been shown to have beneficial therapeutic potential in human health, as evidenced by numerous research findings. Cell culture studies and animal models have revealed that the black seed and its active component TQ have an anti-breast cancer activity. For example, cell metabolism assays indicated that TQ hindered the growth of MDA-MB-231 cells while it did not impact normal cell growth. This effect resulted from G1 phase cell cycle arrest and apoptosis, characterized by the loss of mitochondrial membrane integrity [13]. In the MCF-7 cell line, TQ inhibited cancer cell migration and invasion by targeting the *PPARγ* pathway [14]. Moreover, TQ displayed pro-apoptotic and anti-proliferative effects on the MCF-7 breast cancer cell line by impeding anti-apoptotic proteins like Bcl-2, survivin, and Bcl-xL [15]. However, consistent with earlier studies, our results supported the use of pure TQ for treating cancer [16,17]. TQ-treated animals exhibited a delay in tumor growth compared to untreated tumor-bearing mice. This aligned with earlier studies demonstrating TQ’s ability to inhibit tumor growth in breast, liver, prostate, and colorectal cancers [18,19]. For instance, Woo et al. studied the role of TQ for preventing human breast cancer using female nude mice (BALB/c OlaHsd-foxn1). The findings of this study revealed that TQ’s induction effect on p38 and ROS signaling mediated both its anti-proliferative and pro-apoptotic effects [20]. In a separate in vivo study involving nude athymic female mice with breast cancer, the results indicated that TQ treatment inhibited *eEF-2K* signaling, thereby, inhibiting cell migration, invasion, proliferation, and tumor growth [21].

IF not only reduces the development of cancers in animals but also slows the growth of many types of produced tumors while enhancing their sensitivity to chemotherapy [8]. Alternatively, TQ produces favorable results in vivo and in vitro studies on different cancer cell lines [17]. However, a comprehensive evaluation of the additional effect of an IF and TQ combination for treating cancer, specifically breast cancer, has not been carried out. This combination could offer an attractive therapeutic option to augment conventional anticancer therapies. As a result, the goal of this study was to assess the potential anticancer activity of a novel TQ and IF combination as a treatment for breast cancer implanted in mice.

## 2. Results

### 2.1. Anti-Proliferative Assay

Different cell lines were treated with increasing concentrations of TQ (3.9–500 µM) compared to increasing concentrations of TQ (3.9–500 µM) in a glucose/serum-restricted medium, and a dose-dependent inhibition of cell proliferation was observed (Figure 1A–D). The combination of TQ and a glucose/serum-restricted medium was more cytotoxic against T47D cells, with IC_50_ values of 48.65 ± 0.29 (μg/mL) (Table 1). On the other hand, the TQ and glucose/serum-restricted medium exhibited a lower activity on MDA-MB-231 cells with IC_50_ values of 59 ± 3.2 (μg/mL). For the VERO cell line, the cytotoxicity of the combination was limited, with IC_50_ values of more than 499.35 ± 8.9 (μg/mL) (Table 1).

### 2.2. Antitumor Effects of Single TQ, IF, and Their Combination on EMT6/P Cells Implanted in Mice

Based on the findings from the in vitro assay, the antitumor effects of TQ, IF, and their combinations were examined in female Balb/C mice. The findings presented in Table 2 show that, compared to the negative control which had an increased tumor size, the treated groups had a significant decrease (*p* < 0.05) in tumor size (Figure 2 and Figure 3). In comparison to the untreated mice (+88.54%), the tumor-bearing mice treated with 10 mg/kg/day of TQ experienced a significant decrease (*p* < 0.05) in tumor growth. The percentage change in the tumor size was −37.74. In the tumor-bearing mice treated with IF, a change in the tumor size of −45% was noted along with a greater inhibition of tumor growth. Even though the combined treatment reduced the tumor size by −57.78%, this reduction was greater than what was seen with the individual treatments.

### 2.3. Effect of Single TQ, IF, and Their Combination on the Glucose and β-Hydroxybutyrate Levels

The subsequent analysis of the glucose levels revealed that IF treatments, either alone or in combination with TQ, resulted in the lowest level of glucose. The animals receiving IF generally had lower glucose levels compared to those receiving TQ as a single treatment or the control. However, this reduction was not significant (*p* > 0.05) (Figure 4).

When examining the impact of the different treatment groups on the serum β-Hydroxybutyrate (β-HB) levels, IF and its combination generated the highest levels of β-HB, demonstrating a statistically significant difference (*p* < 0.05) compared to the other treatments (Figure 5).

### 2.4. Effect of Single TQ, IF, and Their Combination on the Serum Levels of IGF-1

The IGF-1 serum levels were assessed in each treatment group. The group receiving IF alone or in combination with TQ had the lowest serum IGF-1 levels for all the tumor-bearing mice (*p* < 0.05) (Figure 6).

### 2.5. Effect of Single TQ, IF, and Their Combination on the Serum Leptin Concentration

The leptin levels were examined, revealing that the group subjected to IF, whether applied independently or in conjunction with TQ, exhibited the lowest levels of leptin (*p*-value < 0.05). In general, the groups getting IF had lower leptin levels than those receiving TQ alone or the control. On the other hand, when we compared the groups that were treated with IF alone or IF with TQ and with the healthy mice group, the reduction was insignificant (Figure 7).

### 2.6. Effects of Single TQ, IF, and Their Combination on the Serum Levels of AST, ALT and Creatinine

ALT and AST assays were used as indicators for liver toxicity. The serum levels of AST, ALT and creatinine (Cr) were assessed in each group that received treatment. When compared side by side with the levels recorded for the normal, untreated mice, the serum ALT levels for all the treated groups were found to be within the normal range with negligible differences (*p* > 0.05). Additionally, the serum AST levels of all the treatment groups were within the normal range when compared to the group of healthy, normal mice (*p* > 0.05).

Normal levels of serum creatinine were found between the healthy mice and the different treatment groups; however, a single TQ treatment significantly increased the creatinine levels (14.82 mg/dL, *p* < 0.05). It is important to note that combining TQ and IF resulted in lower creatinine levels than TQ alone, with a value of 8.84 mg/dL (Table 3)

## 3. Discussion

Despite the availability of effective treatments, breast cancer remains the most common type of tumor in women. Furthermore, breast cancer has risen to become the second highest cause of death and is still a substantial cause of morbidity and mortality [1,2]. Chemotherapeutic treatments target cancer cells that proliferate aggressively, but they also harm healthy cells. Despite the massive advances in chemotherapy, patient survival rates remain substandard, owing to medication resistance after initial chemotherapy treatments, which halts the progress of the patient’s prognosis coupled with the elevated risk of cancer recurrence [22]. As a result, oncological research is focusing its efforts on developing novel and effective therapies that can prevent or minimize the effects of conventional treatments.

Our research may be the first to investigate the complementary anticancer effects of the natural compound TQ and IF, which have been shown to contribute to a reduction in the risk of breast cancer [23,24,25]. TQ, IF, and their combinations were tested for anticancer activity in vitro and in vivo. The combination of TQ and IF had strong regulatory effects on both the development and growth of breast cancer in vivo and the inhibition of cell growth in vitro. The attempt to improve TQ and IF’s anticancer activity has thus yielded encouraging results, indicating an additive anticancer activity without discernible toxicity. TQ effectively inhibited cell growth in breast cancer cell lines, namely EMT6/P, MDA-231, and T47D, in a dose-dependent manner. Similar responses were observed in human breast adenocarcinoma after exposure to TQ.

In our study, EMT-6/P appeared comparatively more resistant to TQ treatment with an IC_50_ of 131 ± 5 µM, as outlined in Table 1. This resistance could potentially have been attributed to the cell line’s capability to enhance the control of various intracellular pathways associated with cell proliferation and survival, as indicated by the previous research [26]. Such resistant cells produced high amounts of substances that promoted cell progression and transformation [27]. Conversely, when EMT-6/P was exposed to TQ in combination with a glucose-restricted medium, the IC_50_ value showed a lower level, demonstrating a percentage reduction of −58%. This reduction may have been linked to IF and the subsequent decrease in insulin-like growth factor-1 (IGF-1).

TQ is primarily derived from *Nigella sativa*, with the volatile oil of this plant constituting 0.40–0.45%. TQ has been recognized as the primary bioactive element responsible for the health advantages found in this plant [28,29]. In our study, the intraperitoneal administration of 10 mg/kg of TQ resulted in a significant anticancer effect.

The outcomes of this study further affirmed the efficacy of TQ, characterized by minimal alteration in tumor growth, the absence of fatalities, and a significant number of successfully treated mice. In the in vitro and in vivo studies, cell growth was both significantly inhibited by IF and glucose/serum restriction when used alone. Our results showed that glucose restriction led to a slight reduction in the survival percentage. It had remarkably similar effects on EMT6/P, MDA-MB-231, and T47D, with survival rates of 76.44%, 80.33%, and 82.57%, respectively, while the toxicity on the Vero cell line was minimal (Figure 1E). These results were consistent with the earlier research that demonstrated the ability of glucose/serum restriction to inhibit breast cancer cell proliferation in vitro. The impact of restricting glucose was also tested on the MDA-MB 231 and MCF-7 breast cancer cell lines. The results indicated that limiting glucose effectively reduced the Warburg effect by suppressing *mTOR* and regulating embryonic isoform 2 of pyruvate kinase (PKM2) [30]. The “Warburg effect”, recognized as aerobic glycolysis in cancer cells, stands as one of several defining characteristics of these cells. The conversion of phosphoenolpyruvate (PEP) to pyruvate can be prevented by re-expressing PKM2. Through this process, glycolytic intermediates can build up and be used to create macromolecules such as proteins, lipids, and nucleic acids. It is important to mention that within the well-known proliferative PI3K/Akt/mTOR pathway, PKM2 is favored. This pathway is triggered by elevated glucose levels, and the mTOR kinase serves as the primary activator of the Warburg effect [31].

Furthermore, the effect of varying glucose concentrations on the potential anti-proliferative impacts of phenformin on the MCF-7, T-47D, and MDA-MB-231 cell lines was investigated. The results revealed that in diverse molecular subtypes of breast cancer, depriving the cells of glucose heightened the anti-proliferative effects of oral hypoglycemic biguanides. This phenomenon could be attributed to the reduced presence of *AMPK* in the low-glucose environment, leading to decreased *AMPK* phosphorylation upon treatment with phenformin compared to the control [32]. Additionally, glucose restriction resulted in heightened mitochondrial ROS levels and the elevation of the pleiotropic protein Prohibitin-1 (PHB1), causing its detachment from Dynamin-related protein-1 (DRP1). This led to disruptions in the mitochondrial membrane potential (MMP) and initiated the cascade leading to apoptosis [33].

According to our in vitro findings, glucose/serum restriction inhibited breast cancer cells. To improve this capacity, we created an in vivo experiment to examine the impact of TQ alone and in combination with IF on breast cancer cells injected into the mice. Our results showed that the tumor size was reduced in the fasting mice compared to the untreated mice (−45.01%), while the mice treated with IF and TQ exhibited a further reduction in the tumor size (−57.78%) (Table 2). Our findings were consistent with the prior research, showing IF’s ability to inhibit tumor growth in breast cancer [34], liver [35], prostate [36], and colorectal cancers [37].

To gain a deeper understanding of how IF exerted its anticancer impact on the treated mice, we analyzed the glucose, β-HB, IGF-1, and leptin levels in their serum. With low glucose during IF, the body turned to fat for energy, leading to the release of ketone bodies such as acetone and β-HB [38,39]. In our investigation, the cohorts undergoing combined therapy with intermittent fasting showcased diminished blood glucose levels alongside elevated β-HB concentrations. Our suggestion was that the anticancer effects of IF led to a reversal of the Warburg effect, consequently increasing the serum β-HB levels. It is crucial to note the inherent connection between the Warburg effect and cancer. Even under sufficient oxygenation, cancer cells display heightened glycolysis, resulting in increased lactate production and accumulation. This heightened glycolytic activity stimulates the pentose phosphate pathway, generating two molecules of nicotinamide adenine dinucleotide phosphate (NADPH), which are crucial for mitigating oxidative stress within cancer cells [40]. Physiologically relevant doses of β-HB effectively restrained the growth of various human glioblastoma cell lines and a murine glioma cell line in vitro, despite high glucose levels, as demonstrated by earlier studies [41,42]. In our study, the levels of β-HB were increased using IF (Figure 5). Additionally, our results were in agreement with the earlier research that showed comparable IF effects against different cancer types [43,44].

Moreover, our study showed that in all the tumor-bearing mice, there was a significant reduction in the IGF-1 levels in the groups treated with IF and combination therapy (*p*-value < 0.05) (Figure 6). The IGF-1 protein has been most closely associated with breast cancer progression because it has mitogenic and anti-apoptotic effects on mammary epithelial cells. IGF-1 stands out as a key candidate for evaluating therapies that aim to inhibit apoptosis, given its crucial involvement in the apoptosis mechanism. IGF-1 is a growth factor that responds to nutrients and activates two main signaling cascades. The triggering of the Ras/MAPK pathway encourages the activity of transcription factors and the subsequent expression of the genes included in proliferation and cellular growth. The PI3K/AKT pathway augments reduced apoptosis by interfering with the Bcl2-related death promoter (*Bad*) complex. Overall, the two pathways encourage cell proliferation [45]. The AMP-activated protein kinase (AMPK) functions as a molecular sensor in cancer cells, encouraging catabolism and discouraging anabolism. The reduced IGF-1 levels by IF resulted in a reduction in tumor growth and progression because the cancer cells used the IGF-1 signaling pathway to convert their metabolic resources toward proliferation and growth [46]. Our results supported the earlier findings in this regard. For instance, the MCF7 and T47D hormone-receptor-positive breast cancer cell lines were implanted into female athymic Nude-FoxN1 mice. In this study, reducing the circulating levels of IGF-1, insulin, and leptin, periodic fasting, or a diet that mimics fasting enhanced the effects of the endocrine therapeutics tamoxifen and fulvestrant [47].

To gain deeper insight into the mechanism behind intermittent fasting’s anticancer effects, we assessed the leptin levels across all the groups of mice. This hormone, leptin, significantly contributes to breast cancer development by enhancing the activation of the signaling pathways associated with cell proliferation while impeding the apoptotic response. For example, the interaction between leptin and IL-1 in breast cancer cells is linked to the promotion of pro-inflammatory and pro-angiogenic signals, contributing to the progression of breast cancer [48]. Additionally, leptin promotes the production of IL-18 and IL-8, which interfere with breast cancer cells and M2 tumor-associated macrophages, and thus stimulates tumor growth [49]. Reactive oxygen species (ROS) are also stimulated by leptin in human epithelial mammary cells [50]. As a result, developing a new therapeutic strategy to reduce leptin levels in breast cancer is critical. Our study showed a significant reduction in leptin levels in the fasting mice group and the combination therapy group compared to the untreated group (*p*-value < 0.05) (Figure 7). Similarly, a study was done to find out how leptin might impact the activity of breast cancer stem cells (BCSCs) using patient-derived samples. The results showed that mammospheres (aggregates of mammary epithelial stem cells) were significantly less likely to form in breast cancer cell lines and the patient-derived samples when leptin signaling was inhibited by a full leptin receptor antagonist [51]. In addition, in murine mammary tumor virus (MMTV)-Wnt-1 transgenic mice, a leptin deficiency was connected to low BCSC levels [52].

The safety profile of anticancer agents is routinely assessed to determine their toxicity. Liver enzymes like ALT and AST serve as markers for liver function, while creatinine indicates renal function [53]. In our study, we compared the results of the treated groups to those of the normal/healthy untreated group to gauge these parameters. All the treatments appeared to have an acceptable safety profile because the ALT and AST values of the treated group were within normal ranges and lower than those of the control group. No significant disparity in the ALT and AST levels was observed between the combination therapy group and the healthy mouse group (*p*-value > 0.05). Conversely, the mice treated with TQ exhibited the highest creatinine levels, whereas those in the combination therapy group displayed the lowest levels. Consequently, the creatinine levels were lower than those associated with either TQ or IF alone, pointing toward a favorable safety profile (Table 3).

## 4. Materials and Methods

### 4.1. In Vitro Experiments

#### 4.1.1. Cell Culture

To evaluate the anticancer activity of TQ and glucose/serum restriction, human breast cancer cell lines were used, namelyMDA-MB-231, which is the most commonly used triple-negative human breast cancer cell line, T47D, which is an estrogen receptor-positive cell line, and EMT6/P, which is a triple-negative mouse mammary carcinoma cell line. Kidney epithelial cells from the Vero cell line of an African green monkey were used as a reference control. The European Collection of Authenticated Cell Cultures (ECACC; Salisbury, UK) provided all the cell lines used in the experiments. The ideal cell culturing conditions were considered for cell growth using complete tissue culture media and an appropriate incubation environment, including 37 °C, 5% CO_2_, and 95% humidity. To make a complete tissue culture medium, a 10% fetal bovine serum, 1% L-glutamine, 1% Penicillin-Streptomycin solution, 0.1% Gentamycin sulfate solution, and 0.1% non-essential amino acids supplement were added to the media [16]. The MDA-MB-231 cell line was cultured in a complete Dulbecco’s Modified Eagle Medium (DMEM), while the T47D cell line was cultured in a complete Roswell Park Memorial Institute (RPMI) 1640 medium. A Gibco minimum essential medium (MEM) was used to cultivate the Vero and EMT6 cell lines.

#### 4.1.2. Preparation of the Thymoquinone Working Solutions and Glucose/Serum-Restricted Tissue Culture Media

TQ (≥98%) was purchased from Chem Cruz (Dallas, TX, USA) TQ (10 mg) was dissolved in 20 µL of DMSO and 800 µL of a complete tissue culture medium, such as the RPMI 1640 media (2 g/L glucose + 10% FBS), MEM (2 g/L glucose + 10% FBS), and DMEM (4.5 g/L glucose + 10% FBS), to generate a solution of 10 mg/mL known as a stock solution. The process was repeated where 10 mg of TQ was dissolved in 20 µL of DMSO and 800 µL of a glucose and serum-restricted medium, such as a low-glucose DMEM (1 g/L + 1% FBS), free glucose RPMI supplemented with glucose (1 g/L + 1% FBS), and low-glucose (1 g/L + 1% FBS) MEM [54]. Further dilution was made on the stock solutions to prepare the desired concentration of TQ (500 µM) by taking 16 µL from the stock solutions and dissolving it in 1 mL of a complete medium and a glucose/serum-restricted medium. For the in vivo study, TQ was diluted using PBS.

#### 4.1.3. Anti-Proliferative Assay

Cells were seeded into a 96-well tissue culture plate (100 µL/well) using a multi-channel pipette at an established concentration of 15,000 cells/well following cultivation, trypsinization, and cell counting for all cell lines. The plate was incubated for 24 h to assure the adhesion of cells to the wells and proper proliferation. After incubation, the media from the wells was removed, and the attached cells were treated with TQ (3.9–500) µM at increasing concentrations in a complete cell culture medium. The T47D cell lines were cultivated in RPMI 1640 media (2 g/L glucose + 10% FBS), while the EMT6/p and Vero’s cells were cultured in a complete minimum essential medium (MEM) (2 g/L glucose + 10% FBS) and the MDA-MB231 cells were cultured in high glucose (DMEM) (4.5 g/L glucose + 10% FBS). Additionally, the attached cells were also treated by increasing the concentrations of TQ in the glucose and serum-restricted medium as low-glucose (1 g/L + 1% FBS) DMEM, free glucose RPMI supplemented with glucose (1 g/L + 1% FBS), and low-glucose (1 g/L + 1% FBS) MEM [54].

The effects of TQ and glucose/serum restriction on breast cancer cell viability were assessed using the MTT assay. This assay was based on the detection of blue formazan crystals that form when mitochondrial dehydrogenase reduces MTT dye. The presence of formazan blue crystals suggests that mitochondrial function is normal, and thus precludes cell survival. After fully removing the drug solution from each well, each well was refilled with 100 µL of a fresh complete tissue culture medium and 20 µL of an MTT solution. The plates were incubated for three hours. To stop the reaction, 100 µL of DMSO was added as an MTT solubilizing agent. The plates were returned to the incubator for another hour. A microplate reader (Biotek, Winooski, VT, USA) was used to measure the absorbance at 550 nm. In comparison to the negative control cells, which only contained tissue culture media, the percentage of cell viability was calculated. The cells were exposed for different treatments for 48 h. The percentage of survival was calculated using the following equation. Cell viability = (Sample abs/control abs) × 100%. Each experiment was repeated three times.

### 4.2. In Vivo Experiments

All the experimental procedures were approved by the Research and Ethical Committee of the Faculty of Pharmacy, Applied Science University and were conducted under standard ethical principles. The number of ethical approvals was 2021-PHA-17.

In this study, 27 healthy female Balb/C mice were used. They ranged in weight from 21 to 25 g and were 6 to 8 weeks old. The mice were placed in well-ventilated rooms with a room temperature of 25 °C, 50–60% humidity, and alternate dark and light cycles every 12 h to establish the required environmental conditions. They were housed in cages with bedding made of wood shavings, a special water bottle, and a place to eat.

#### 4.2.1. Establishing the TQ Dose for the In Vivo Experiment

Female Balb/C mice bearing EMT-6/P tumors were used for the in vivo research. A dose was determined using the reported doses from the literature. Considering the results of an earlier study, the TQ in vivo dose of 10 mg/kg/day was chosen [24].

#### 4.2.2. Tumor Inoculation and Antitumor Activity Assay

EMT-6/P cells were taken from the nitrogen tank, thawed, and cultured in 75 cm^2^ flasks containing 15 mL of MEM. Under the required conditions, the cells were incubated. Using trypsin-EDTA, the cells were detached from the flask wall once they had actively divided and formed a confluent layer. On day 0 of the in vivo experiment, a suspension of 1.5 million cells/1 mL of MEM was prepared, and a tumorigenic dose of 150,000 cells/100 μL was injected subcutaneously into the female Balb/C mice. The tumors had the chance to grow for 14 days before the tumor-bearing mice began receiving the experimental treatments on day 15 of tumor inoculation.

Three time points during the duration of treatment—days 1, 5, and 10—were used to assess the health, behavior, and tumor volumes of the mice. The following equation was used to determine the percentage of the tumor volume change between the initial and final volumes.
Tumor volume=A × B2 × 0.5
where

A = the length of the tumor’s longest side.

B = the length of the tumor aspect perpendicular to A.

Afterward, four groups of mice were formed. To closely match the average tumor volume in each group, tumors of comparable sizes were selected.

Group I was the TQ group (n = 7 mice), which received an intraperitoneal injection of TQ at a dose of 10 mg/kg/d.

Group II was subjected to intermittent fasting for 18 h every day (n = 7 mice).

Group III was the combination group (n = 7 mice); the mice fasted for 18 h each day and were given 10 mg/kg/d of TQ intraperitoneally.

Group IV was the control group (n = 6 mice) where the mice received daily intraperitoneal injections of the vehicle (2% tween 20 in PBS).

The mice were euthanized by cervical dislocation at the end of the experiment, and the tumors and organs were removed, weighed, and preserved in 10% formalin. There was no need for an earlier termination for any of the mice during the study due to the short treatment period, which was 10 days.

#### 4.2.3. Evaluation of the Serum Levels of Glucose and β-Hydroxybutyrate 

On days 1, 5, and 10, the serum levels of glucose and β-HB were assessed and compared to those of the untreated normal mice. The serum glucose levels were determined using the Accu-Chek blood glucose monitoring system from Roche, Basel, Switzerland. The β serum levels were determined using the ELISA Kit from My BioSource San Diego, CA, USA (Cat. No. MBS2612519).

#### 4.2.4. Measuring Serum IGF-1

The levels of the serum IGF-1 were measured using the IGF-1 Mouse ELISA kit (Invitrogen, Waltham, MA, USA) (Cat. No. EMIGF1X10). In this experiment, a mouse IGF-1-specific antibody was applied to a 96-well plate. The immobilized antibody bound to the IGF-1 present in the samples after being pipetted into the wells with the standards and samples. An anti-mouse IGF-1 antibody that had been biotinylated was added to the wells after washing. The unbound biotinylated antibody was removed and HRP-conjugated streptavidin was pipetted into the wells. After a second round of washing, a TMB substrate solution was added, and the color of the wells changed in direct proportion to the concentration of bound IGF-1. At 450 nm, the color intensity of the stop solution was measured as it changed from blue to yellow.

#### 4.2.5. Measuring the Serum Leptin Concentration

The Leptin Mouse ELISA kit (Invitrogen, Vienna, Austria) (Cat. No. KMC2281) was used to assess the serum leptin. An in vitro enzyme-linked immunosorbent assay was used to measure mouse leptin quantitatively in the serum. In this experiment, an anti-mouse leptin antibody was coated onto a 96-well plate. Leptin from the sample, which was pipetted into the wells along with the standards and samples, was bound by the immobilized antibody. The wells were thoroughly cleaned and an anti-mouse leptin antibody was added. After removing the unbound biotinylated antibodies with a washing step, HRP-conjugated streptavidin was pipetted into the wells. A TMB substrate solution was then added, and the color developed in direct proportion to the amount of leptin that was bound. The wells were then rinsed once more. At 450 nm, the color intensity of the stop solution was measured as it changed from blue to yellow.

#### 4.2.6. Evaluation of Liver and Kidney Functions in the Treated Mice

Kidney and liver toxicity after the application of IF, TQ, and their combination were assessed by measuring the serum creatinine levels and the levels of aspartate aminotransferase (AST) and alanine aminotransferase (ALT), respectively. After the serum samples were collected, the ALT and AST levels were assessed using the ALT/GPT and AST/GOT kits (BioSystems, Barcelona, Spain), respectively. An AU480 chemistry analyzer (Bechman Coulter Brea, CA, USA) was used to measure the creatinine levels.

### 4.3. Statistical Analysis

The data from three separate experiments were presented using the mean and SEM. To determine the statistical significance between the groups, an analysis of variance (ANOVA) and the Tukey test were used (*p* < 0.05 was regarded as significant). The IC_50_ values of TQ and against the various cell lines were determined using nonlinear regression in the Statistical Package for the Social Sciences version 25 (SPSS Inc. Chicago, IL, USA).

## 5. Conclusions

In summary, the combination of TQ and IF presented an intriguing therapeutic approach for the treatment of breast cancer. This combination demonstrated anticancer properties by reversing the Warburg effect, primarily through the reduction in blood sugar levels. Additionally, it elevated the serum levels of β-HB while significantly decreasing IGF-1 and leptin. These combined effects led to a notable reduction in the tumor size in the treated mice. By intervening in the same pathways activated by TQ, it is evident that enhancing TQ’s anticancer activity is achievable. To be considered a potential therapeutic option for breast cancer, further investigation is warranted into this readily available and cost-effective plant-based natural treatment in conjunction with calorie restriction. However, a more comprehensive understanding of the mechanisms of action and its efficacy against other types of cancer requires additional research.

## Figures and Tables

**Figure 1 plants-13-00035-f001:**
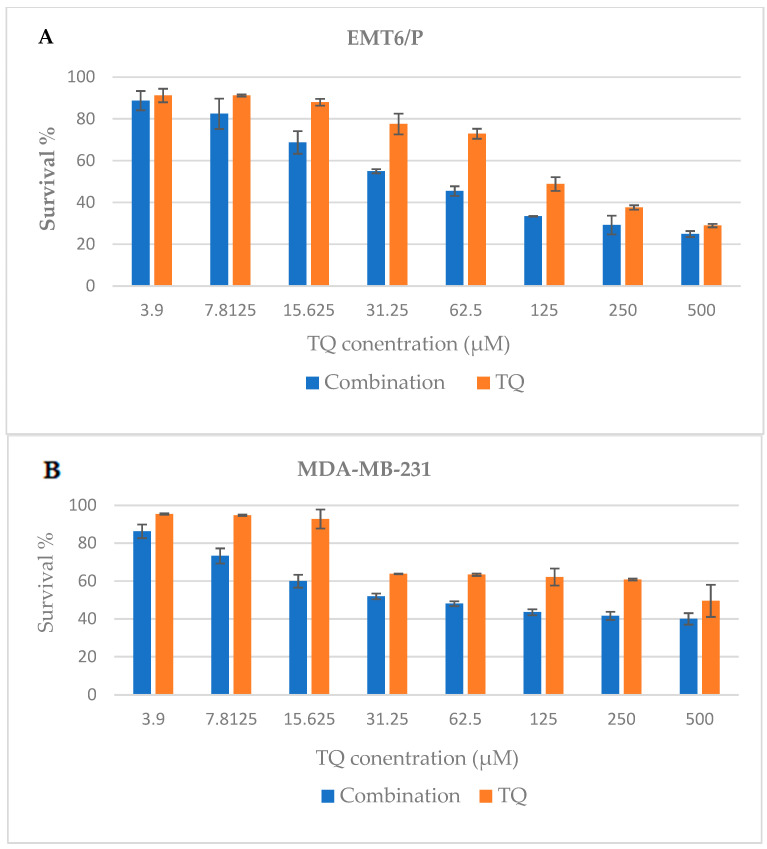
(**A**) Anti-proliferative effect of TQ and the combination treatment (TQ + nutrients restriction) against the EMT-6/P cell lines. (**B**) Anti-proliferative effect of TQ and the combination treatment (TQ + nutrients restriction) against MDA-MB-231. (**C**) Anti-proliferative effect of TQ and the combination treatment (TQ + nutrients restriction) against the T47D cell lines. (**D**) Anti-proliferative effect of TQ and the combination (TQ + nutrients restriction) treatment against the Vero cell lines. In the first three concentrations, the percentage of cell survival was larger than 100%, and perhaps TQ stimulated Vero cell proliferation. (**E**) Anti-proliferative effect of the glucose/serum restriction treatment against the EMT-6/P, MDA-MB-231, T47D and Vero cell lines. The results are expressed as the means ± SEM.

**Figure 2 plants-13-00035-f002:**
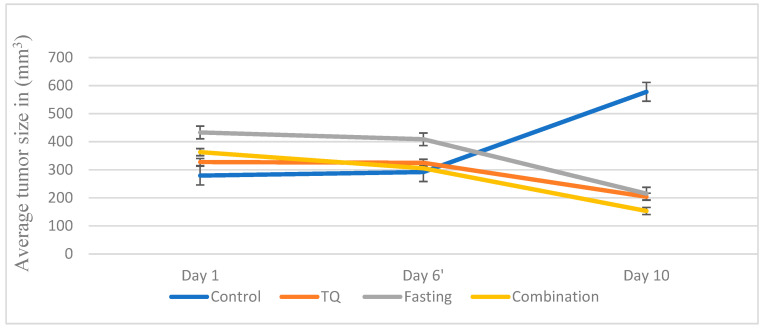
A plot of the change in the average tumor size (mm^3^) over time (in days) for treatment in the EMT-6/Pcell line (*p* < 0.001) compared to the control group.

**Figure 3 plants-13-00035-f003:**
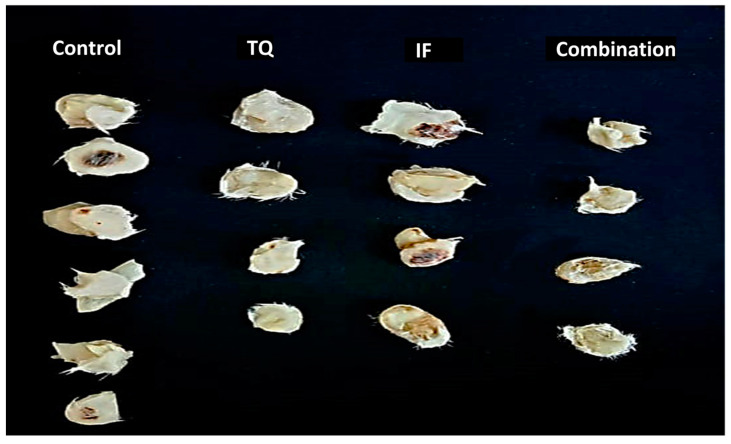
Comparison of the tumor size in each group (single TQ, IF, and their combinations) after day 10 via dissection in EMT-6/P (n = 7) compared to the control group (n = 6).

**Figure 4 plants-13-00035-f004:**
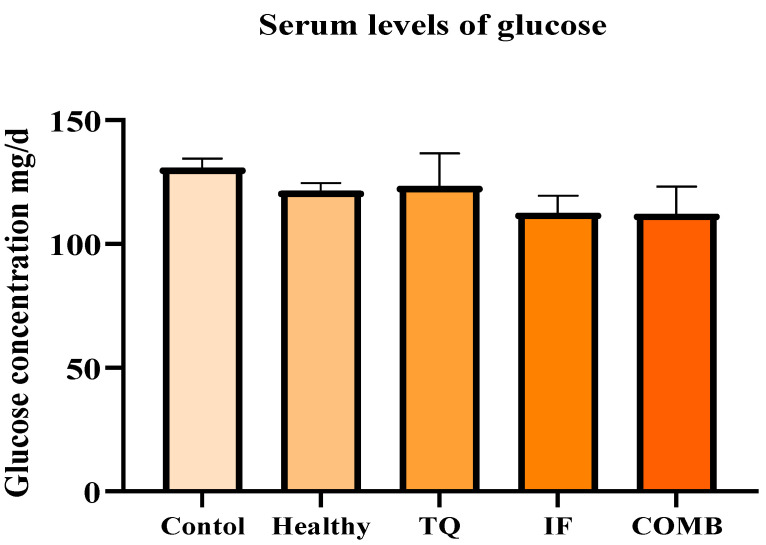
Serum levels of glucose for the different treatment groups. The lowest glucose levels were observed in the IF treatment group either alone or in combination with TQ. The results are represented as the mean ± SEM (n = 3).

**Figure 5 plants-13-00035-f005:**
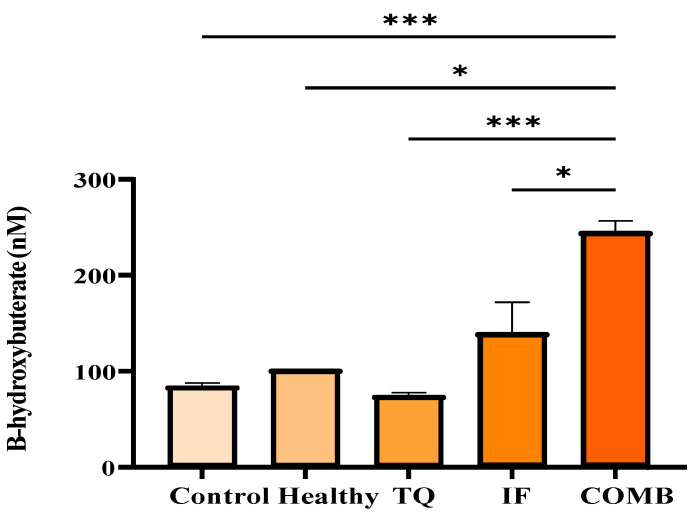
Serum levels of β-HB for the different treatment groups. The highest β-HB levels were observed in the IF treatment group either alone or in combination with TQ. The results are represented as the mean ± SEM. * *p*-value < 0.05, *** *p*-value < 0.005 (n = 5).

**Figure 6 plants-13-00035-f006:**
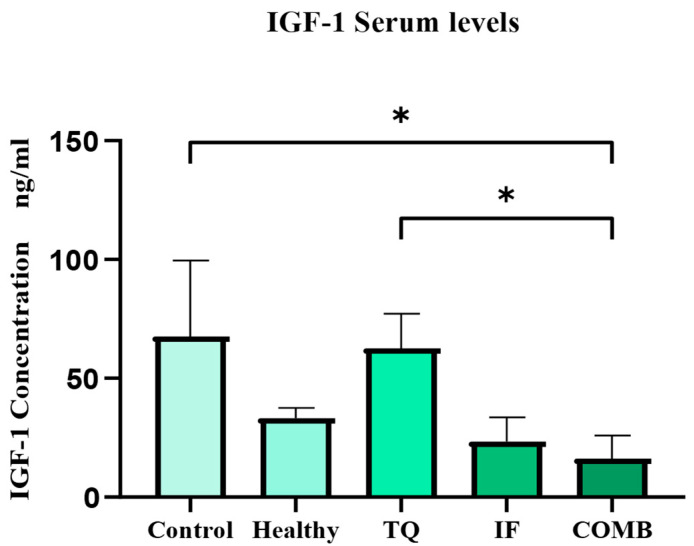
Serum levels of IGF-1 for the different treatment groups. The lowest IGF-1 levels were observed in the IF treatment group either alone or in combination with TQ. The results are represented as the mean ± SEM. * = *p*-value < 0.05 (n = 3).

**Figure 7 plants-13-00035-f007:**
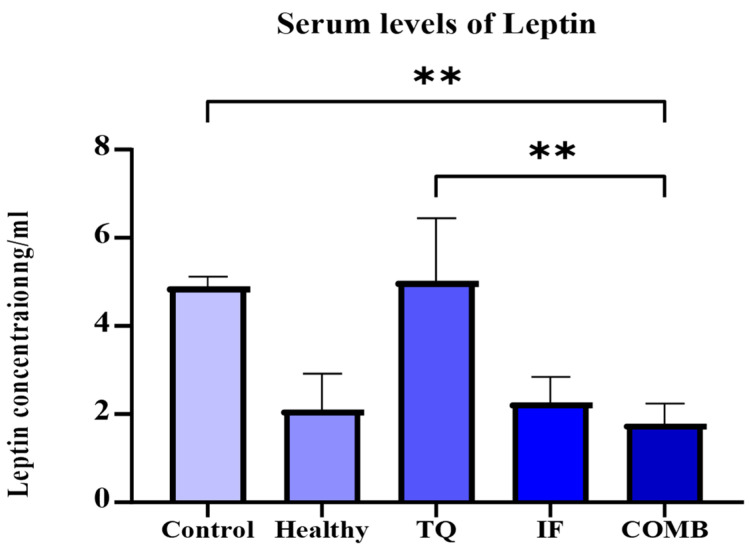
Serum levels of leptin for the different treatment groups. The lowest leptin levels were observed in the IF treatment group either alone or in combination with TQ. The results are represented as the mean ± SEM. ** = *p*-value < 0.05 (n = 3).

**Table 1 plants-13-00035-t001:** IC_50_ (μM) for the single TQ and combination treatments.

Cell Line	EMT-6 IC_50_ (μM) ± SEM	MDA-MB-231 IC_50_ (μM) ± SEM	T47D IC_50_ (μM) ± SEM	Vero IC_50_ (μM) ± SEM
Single TQ	131 ± 5.0	100 ± 5.5	106 ± 5.7	497.7 ± 1.6
Combination	55 ± 4.0	59 ± 3.2	48.65 ± 0.29	499.35 ± 8.9

**Table 2 plants-13-00035-t002:** Effect of single TQ, IF, and their combinations in the EMT-6/P cell line (n = 7) on the average tumor weight, percentages of change in the tumor size, and tumor size changes compared to the control group (n = 6).

	Av. Initial Tumor Size (mm^3^) ± SEM	Av. Final Tumor Size (mm^3^) ± SEM	% Change in Tumor Size	% of Mice with No Detectable Tumor	Average Tumor Weight (g)
TQ	327.7 ± 24.7	204 ± 12.6	−37.74	42.85714	0.362 ± 0.05
IF	391.4 ± 20	215.24 ± 22.4	−45.01	42.85714	0.316 ± 0.12
Combination of TQ and IF	363 ± 17	153.28 ± 5.23	−57.78	42.85714	0.298 ± 0.088
Control	306.6 ± 17.5	578.04 ± 62.43	88.54	0	0.45 ± 0.63

**Table 3 plants-13-00035-t003:** Serum ALT, AST, and Cr levels for the different treatment groups, control, and normal untreated mice.

Treatment Groups	ALT (IU/L)	AST (IU/L)	Cr (µmol/L)
TQ	37.774 ± 4.44	82.21 ± 9.44	14.82 ± 0.0866
IF	38.32 ± 2.77	117.76 ± 4.44	13.26 ± 0.0866
Combination	45.55 ± 2.22	107.21 ± 10.55	8.84 ± 0.0866
Control	75.33 ± 1.29	101.65 ± 10.55	15.91 ± 0.0866
Healthy mice	47.77 ± 4.713	117.76 ± 4.44	8.84 ± 0.0866

## Data Availability

The data are available within the article.

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
