# Peer review of "Combination of Thymoquinone and Intermittent Fasting as a Treatment for Breast Cancer Implanted in Mice"

_plants, 2023, doi:10.3390/plants13010035_

Round 1

Reviewer 1 Report

Comments and Suggestions for Authors

Haif et al. evaluated the combined effects of Thymoquinone and Intermittent Fasting on breast cancer treatment through both in-vitro and in-vivo studies. The manuscript is notable for its clear and professional use of the English language, as well as well-organized tables. The authors have also written a comprehensive discussion section that thoroughly examines the factors influencing cancer cell cytotoxicity and tumor suppression. Furthermore, the manuscript includes a review of up-to-date literature and a current list of references. My comments primarily focus on the figures, with the most critical issue being the absence of Figure 3 in the reviewer's copy of the manuscript. Please find my detailed comments below.

(1) Figure 1D: Even considering the error bars, in the first three concentrations, the percentage of cell survival is larger than 100%. I would recommend adding a comment in the main text or figure captions explaining why this is happening (perhaps Thymoquinone was facilitating Vero cell growth), to prevent confusion among readers.

(2) Figure 1E: The x-axis label is missing, making it difficult for readers to understand the figure by just looking at it.

(3) (Major Point): Figure 3 is missing in the reviewer's copy of the manuscript.

Author Response

Thank you for your comments. Please see our response attached

Reviewer 2 Report

Comments and Suggestions for Authors

In general, the paper should be shortened in all its parts, to make it easier to read. The discussion paragraph, in particular, is far too long and redundant. Many concepts, for example those exposed in lines 284-291 and 305-316, should be more conveniently moved to the introduction, also to provide a rationale for the choice of breast cancer cell lines and experimental design. Furthermore, greater caution is appropriate as far as the conclusions are concerned.

In addition:

-What is the rationale behind the choice of tumor cell lines?

-it is necessary to indicate (i) the duration of the treatment of the in vitro experiments, (ii) how the percentage of viability was calculated, and (iii) the number of technical and biological replicates.

-Has the effect of nutritional deprivation alone on the growth/proliferation of the different tumor lines been considered?

-Has the effect of the different doubling times of the lines used been considered?  

-The results of the in vitro experiments could be more conveniently represented in a single figure.

- IC50 should be expressed in µM

-Regarding the in vivo experiments: why was the i.p. implant of tumor cells chosen? A representative picture of tumor growth in control and treated mice should be provided.

Comments on the Quality of English Language

No specific comments.

Author Response

Thank you for your comments, Please see attached our response

Round 2

Reviewer 1 Report

Comments and Suggestions for Authors

The authors have addressed my comments and added Figure 3 back to the manuscript. However, I have some additional comments regarding the newly added Figure 3:

In Figure 3 caption, I understand why it is N = 6 for the control group, but where is the N = 7 from? I could not find an experimental group that has a sample size of N = 7.

Also, just by looking at Figure 3, it is hard for readers to tell that "the treated groups had a significant decrease in tumor size." I could only discern that the tumor sizes in the combined treatment group seem to be smaller by eyeballing the tumor images, but the tumor sizes in the other treatment groups and the control group seem to be similar. The authors should add a scale or other quantitative data to Figure 3 to support their statement that "the treated groups had a significant decrease in tumor size."

Reviewer 2 Report

Comments and Suggestions for Authors

The paper has been appreciably improved.

Comments on the Quality of English Language

More concise language would improve reading.